# Orbital-selective band engineering realizes high $zT$ in $p$-type Ru$_2$Ti$_{1-x}$Hf$_x$Si full-Heusler thermoelectrics

Fabian Garmroudi [1,6] ✉, Illia Serhiienko [2,6], Michael Parzer [3], Andrej Pustogow [3], Raimund Podloucky [4], Takao Mori [2,5] & Ernst Bauer [3] ✉

Heusler compounds have emerged as important thermoelectric materials due to their combination of promising electronic transport properties, mechanical robustness and chemical stability – key aspects for practical device integration. While a wide range of XYZ-type half-Heusler compounds have been studied for high-temperature applications, X$_2$YZ-type full-Heuslers, often characterized by narrower band gaps, may offer potential advantages at different temperature regimes but remain less explored. In this work, the discovery of $p$-type Ru$_2$Ti$_{1-x}$Hf$_x$Si full-Heusler thermoelectrics, exhibiting a high figure of merit $zT = 0.7$ over a broad range of temperatures 700–1000 K, is reported. These results not only represent the largest values known to date among full-Heusler materials but confirm earlier theoretical predictions that $p$-type Ru$_2$TiSi systems would be superior to their $n$-type counterparts. Moreover, using a two-band model, electronic structure changes induced by the Hf substitution at the Ti site are unveiled and strategies to further improve $zT$ up to $zT > 1$ are outlined. These findings highlight the untapped potential of new semiconducting full-Heusler phases and the crucial need for continued exploration of this rich materials class for thermoelectric applications.

Thermoelectric (TE) materials exploit the Seebeck effect to generate an electrical voltage from a temperature gradient. This principle offers significant potential for converting waste heat—abundantly produced in industrial processes and typically dissipated into the environment—into usable electrical energy, thereby contributing to the development of more sustainable and energy-efficient technologies[1]. The dimensionless, material-dependent figure of merit, $zT = S^2\rho^{-1}\kappa^{-1}T$, determines the efficiency of such conversion processes and depends on the absolute temperature $T$, the Seebeck coefficient $S$, the electrical resistivity $\rho$ and the thermal conductivity $\kappa$. Due to the interdependence of these physical parameters, improving $zT$ presents a complex and ongoing materials design challenge[2,3]. Since the discovery of the first TE semiconductors in the mid-20th century[4–6],

several high-performance TEs have been developed from various semiconducting material families, such as Pb- and Sn-based chalcogenides[7–11], skutterudites[12–15], clathrates[16–18], various Zintl phases[19–23], and Heusler compounds[24–27] and recently also metallic materials[28–33]. Compared to other semiconducting materials, Heusler systems, being the subject of the current study, prevail in terms of mechanical strength[34], chemical and thermodynamic long-term stability and cost effectiveness – crucial attributes for the development of robust and durable TE modules that are suitable for a variety of practical applications.

Heusler compounds are a class of cubic intermetallics broadly categorized into half-Heusler (hH) phases with XYZ stoichiometry and full-Heusler (fH) phases with X$_2$YZ stoichiometry, where X and Y are

[1]Materials Physics Applications – Quantum, Los Alamos National Laboratory, Los Alamos, NM, USA. [2]Research Center for Materials Nanoarchitectonics (MANA), National Institute for Materials Science (NIMS), Tsukuba, Japan. [3]Institute of Solid State Physics, TU Wien, Vienna, Austria. [4]Institute of Materials Chemistry, Universität Wien, Vienna, Austria. [5]Graduate School of Pure and Applied Sciences, University of Tsukuba, Tsukuba, Japan. [6]These authors contributed equally: Fabian Garmroudi, Illia Serhiienko. ✉e-mail: fgarmroudi@lanl.gov; bauer@ifp.tuwien.ac.at

typically transition metals and Z is a main group element from groups III to V[35]. Their chemical and electronic properties follow simple electron-counting rules, such as the Slater-Pauling principle. This enables the rational design of semiconducting ground states—particularly relevant for TE applications—by targeting an average valence electron count of six per atom (VEC = 6)[35–37]. Notably, hH compounds with VEC = 6 tend to exhibit wider band gaps than their fH counterparts, which can be attributed to the reduced symmetry and $d$–$d$ hybridization. In fH compounds, the additional X atom introduces strong $d$–$d$ hybridization that broadens the valence and conduction manifolds, often narrowing or even closing the band gap. As a result, hH systems are generally better suited for high-temperature TE applications[25,38–41], since optimal TE performance is often achieved when the temperature reaches a fraction of the band gap $E_g \sim 10\,k_B T_{work}$, where $k_B$ is the Boltzmann constant and $T_{work}$ denotes the working temperature for optimal device operation[42].

Among fH compounds with promising electronic structures, experimental efforts have predominantly focused on Fe$_2$VAl-based systems[43–47]. The undoped parent compound is characterized by a narrow pseudogap (or almost-zero band gap) near the Fermi energy $E_F$, accompanied by a steeply rising density of states (DOS) at either side of $E_F$[48,49]. While immense progress with respect to enhancing $zT$ in both $p$-type[50–54] and $n$-type Fe$_2$VAl-based materials[55–61] has been made over the recent years, the maximum figure of merit $zT_{max} \approx 0.3 - 0.4$ of the best-performing systems still falls short by a factor of 2–3 compared to the benchmark material Bi$_2$Te$_3$, currently utilized in commercially available TE modules. Thus, it is crucial to explore other semiconducting fH phases with narrow band gaps that could potentially replace Bi$_2$Te$_3$ as a more robust option for low- to mid-temperature TE applications on the long run. In this context, Fujimoto et al. recently explored Ru$_2$TiSi as a new TE fH material with VEC = 6[62]. Initial investigations into the TE properties of $n$-type Ta-substituted Ru$_2$Ti$_{1-x}$Ta$_x$Si systems revealed a $zT_{max} \sim 0.4$ at high temperatures, around 900 K[62] due to a larger band gap compared to their Fe$_2$VAl-based relatives. In a subsequent study[63], a detailed two-band model analysis of the temperature and doping dependence of the TE properties showed that the electronic band structure of Ru$_2$TiSi promises a much greater potential for $p$-type materials, if the lattice contribution of the thermal conductivity ($\kappa_L$) could be reduced by isovalent heavy-element substitution, e.g. in Ru$_2$Ti$_{1-x}$Hf$_x$Si. Specifically, a large $zT_{max} > 1$ was theoretically predicted for optimally doped $p$-type Ru$_2$Ti$_{1-x}$Hf$_x$Si at $T = 700$ K, along with an attractive $zT \sim 0.4$ around room temperature[63].

Motivated by these initial findings and the predicted enormous potential, we experimentally investigated the structural and TE properties of Ru$_2$Ti$_{1-x}$Hf$_x$Si as a function of Hf substitution. This study is organized as follows: we begin by examining the solubility limit of Hf in the Ru$_2$Ti$_{1-x}$Hf$_x$Si system and the resulting microstructures across a broad range of Hf concentrations $x$. We then present and analyze the corresponding TE properties. Finally, we apply a parabolic two-band model to elucidate the electronic structure modifications induced by Hf substitution at the Ti site and discuss strategies for further enhancing TE performance through rational co-substitutions.

## Results and Discussion
### Solubility limit and microstructure
Polycrystalline samples with nominal compositions Ru$_2$Ti$_{1-x}$Hf$_x$Si ($x$ = 0, 0.1, 0.2, 0.3, 0.4) were synthesized by arc melting followed by spark plasma sintering (SPS) at NIMS. Powder X-ray diffraction (XRD) patterns, shown in Fig. 1a, confirm that all samples crystallize in the same L2$_1$-ordered fH structure (Fig. 1c). The diffraction peaks can be indexed with the cubic space group $Fm\bar{3}m$, corresponding to the Cu$_2$MnAl prototype, where Ru occupies the 8c Wyckoff position, Ti or Hf occupies 4a, and Si occupies 4b (Fig. 1a). Minor impurity peaks in

the angular range $2\theta = 33°–39°$ (Fig. 1a) are attributed to Ru-rich secondary phases, consistent with previous reports by Fujimoto et al.[62].

The lattice parameter $a$, plotted in Fig. 1b, increases linearly with increasing Hf content for $x \leq 0.2$, in agreement with Vegard's law. This behavior originates from the larger atomic radius of Hf (158 pm) compared to Ti (146 pm)[64], and confirms full solubility of Ti and Hf at the 4a site in this composition range. Rietveld refinements of the XRD diffraction data (Fig. S1 and Table S1) further confirm that both Ti and Hf occupy the 4a site, also revealing that the actual Hf content in the single-phase region closely matches the nominal composition.

For comparison, lattice parameters of independently synthesized samples prepared at TU Wien by high-frequency induction melting are included in Fig. 1b as well (see also Fig. S1 and Table S1). The close agreement between the two data sets confirms the reproducibility of phase formation and lattice expansion across different synthesis methods. For compositions with $x > 0.2$, the lattice parameter deviates from Vegard's law and saturates near $x \sim 0.26$. This deviation coincides with the appearance of an additional fH phase, whose reflections match those of Ru$_2$Ti$_{0.2}$Hf$_{0.8}$Si synthesized separately at TU Wien. We attribute this observation to phase separation in the $x$ = 0.3 and 0.4 samples, resulting in the coexistence of Ti-rich and Hf-rich fH phases. The discontinuity in lattice parameter evolution between $0.26 \leq x \leq 0.8$ indicates a miscibility gap in the Ru$_2$TiSi–Ru$_2$HfSi pseudo-binary system, limiting complete solid solubility between the two end members.

To investigate the phase composition and microstructure, scanning electron microscopy (SEM) was carried out on polished cross-sections. As shown in the backscattered electron (BSE) images in Fig. 2 and the element maps (see Fig. S2), samples with $x \leq 0.2$ exhibit homogeneous compositions, well-sintered grains, and no secondary phases, except for rare minor inclusions (Fig. 2a, c). The elemental maps also reveal slight Hf inhomogeneity within $x \leq 0.2$, which, together with the gradual increase in the FWHM up to $x < 0.3$ (Table S1), suggests that Ru$_2$Ti$_{1-x}$Hf$_x$Si behaves rather like a substitutional alloy. For $x \geq 0.3$ (Fig. 2d, e), multiple phases appear at grain boundaries and within the matrix, confirming structural inhomogeneity as observed also by XRD.

Energy-dispersive X-ray spectroscopy (EDS) reveals pronounced compositional heterogeneity in the $x$ = 0.3 and 0.4 samples. These samples include Hf-rich Ru$_2$Ti$_{1-x}$Hf$_x$Si and a Ru-(Ti,Hf) intermetallic alloy, appearing as gray and white regions, respectively, in Fig. 2d, e. Additionally, a minor fraction of Ru$_2$Ti$_{0.9}$Hf$_{0.1}$Si is detected in the $x$ = 0.4 sample. Although not resolved by XRD, its presence is evident from distinct BSE contrast and EDS measurements, and it is likely undetectable in diffraction due to its lattice parameter being similar to that of the dominant Ru$_2$Ti$_{0.74}$Hf$_{0.26}$Si phase.

In summary, through XRD and SEM analyses, we demonstrate that Ru$_2$Ti$_{1-x}$Hf$_x$Si samples with $x \leq 0.2$ form a single-phase fH structure with uniform microstructure and lattice parameters that follow Vegard's law, indicating complete solubility of Ti and Hf at the 4a site. Beyond this limit, deviations from Vegard's law, additional diffraction peaks, and visible phase separation mark the onset of a miscibility gap for $0.26 \leq x \leq 0.8$. At higher Hf content ($x \geq 0.8$), the system returns to single-phase behavior, with the lattice parameter once again following Vegard's law. The presence of secondary phases in the $x$ = 0.3 and 0.4 samples leads to reduced TE performance. In contrast, the single-phase Ru$_2$Ti$_{1-x}$Hf$_x$Si samples with $x$ = 0.1 and 0.2—within the solubility limit—exhibit promising TE properties, which will be discussed in detail in the following section. These compositions can thus be used as the basis for future co-substitution studies.

### TE properties
Figure 3 shows the temperature-dependent TE properties of Ru$_2$Ti$_{1-x}$Hf$_x$Si for $x$ = 0, 0.1, 0.2, 0.3, and 0.4. All of the samples characterized with respect to their TE properties were synthesized at NIMS as described in Materials and Methods. Upon isovalent substitution of

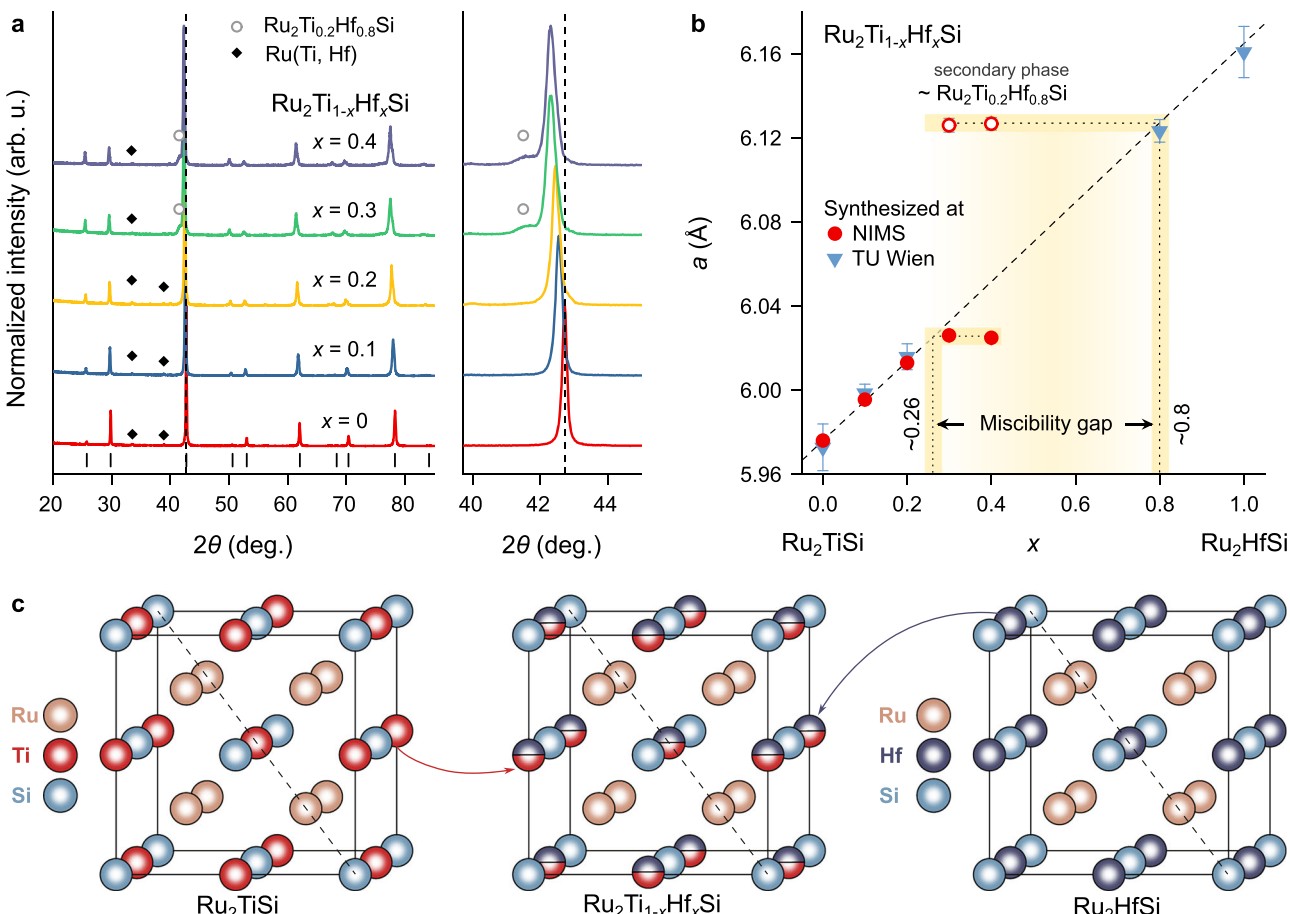

**Fig. 1 | Crystal structure and solubility limit of Hf-substituted Ru$_2$Ti$_{1-x}$Hf$_x$Si full-Heusler compounds. a** X-ray powder diffraction (XRD) patterns of Ru$_2$Ti$_{1-x}$Hf$_x$Si with $x$ = 0, 0.1, 0.2, 0.3, and 0.4. Vertical black ticks at the bottom indicate Bragg reflection positions for the full-Heusler phase with cubic symmetry ($Fm\bar{3}m$). Peaks marked with diamonds and open circles correspond to secondary phases of Ru-(Ti,Hf) and Ru$_2$Ti$_{0.2}$Hf$_{0.8}$Si, respectively. A magnified view of the (220) peak region is shown on the right. **b** Lattice parameter $a$ at room temperature as a function of Hf content ($x$) in Ru$_2$Ti$_{1-x}$Hf$_x$Si synthesized at NIMS (circles) and TU Wien (inverted triangles). Dashed line represents Vegard's approximation. The discontinuity between $x$ ~ 0.26 and 0.8 indicates a miscibility gap. Open circles represent the lattice parameter of the secondary phase. **c** L2$_1$-ordered full-Heusler crystal structure of Ru$_2$TiSi, Ru$_2$HfSi and Ru$_2$Ti$_{1-x}$Hf$_x$Si solid solutions, which adopt the full-Heusler structure only for $x$ < 0.26 and $x$ > 0.8.

Hf for Ti, changes are observed in the temperature-dependent Seebeck coefficient $S(T)$ (Fig. 3a), even for samples confirmed to be within the single-phase regime ($x$ = 0, 0.1, 0.2) by XRD and SEM. This suggests modifications in the electronic structure arising from Hf substitution at the Ti site.

For Ru$_2$TiSi, $S(T)$ reaches a maximum value of approximately 200 µV K$^{-1}$ at around 900 K, in good agreement with earlier reports by Fujimoto et al.[62] and Garmroudi et al.[63]. Upon Hf substitution in Ru$_2$Ti$_{1-x}$Hf$_x$Si, the temperature of the $S(T)$ maximum, $T_{max}^S$, initially decreases with increasing $x$, while the peak value $S_{max}$ remains nearly unchanged. For $x$ > 0.2, a sudden drop in $S(T)$ is observed, coinciding with the emergence of a multi-phase microstructure comprising metallic phases and Hf-rich Ru$_2$Ti$_{1-x}$Hf$_x$Si compositions, which likely exhibit inherently lower $S(T)$. The formation of multiple phases also alters the stoichiometry of the main phase and thereby its carrier concentration, making comparisons with single-phase samples difficult. Moreover, analyzing electronic transport in composite materials can lead to misleading and wrong interpretations of the overall TE performance[65,66]. Therefore, we decided to focus our discussion on the TE properties of the single-phase samples. Figure 3b shows the temperature-dependent electrical resistivity $\rho(T)$. Particularly noteworthy is that $\rho(T)$ does not increase significantly upon Hf substitution and even decreases slightly, in contrast to the pronounced rise observed for $n$-type substitution with Ta at the Ti site[62]. As previously

discussed in ref. 63, this behavior can likely be attributed to the different orbital-decomposed contributions to the electronic structure: the conduction band in Ru$_2$TiSi has predominant Ti $e_g$ orbital character, meaning that disorder introduced at the Ti sublattice leads to strong random potential fluctuations primarily affecting charge carriers in the Ti $e_g$ conduction band states.

In contrast, for Hf-substituted compounds, the chemical potential remains within the Ru $t_{2g}$-dominated valence bands. These bands maintain high conductivity and are less susceptible to impurity scattering caused by disorder at the Ti site (see also the evolution of the concentration-dependent Hall mobility in Fig. S6). On the contrary, substitution at the X site would most likely result in elevated disorder scattering and deteriorated carrier mobility for hole-type carriers[51,67]. Similar trends in electrical resistivity due to X and Y site substitution have also been reported for Fe$_2$VAl[48]. As a result, the electronic performance—as reflected by the power factor $S^2/\rho$ (Fig. 3c) and the weighted mobility $\mu_W$ (Fig. 3d) calculated from the Seebeck coefficient and resistivity via the formula given in ref. 68—does not degrade with increasing Hf content, in contrast to the $n$-type Ru$_2$Ti$_{1-x}$Ta$_x$Si system[62].

The temperature-dependent thermal conductivity $\kappa(T)$ and its lattice contribution $\kappa_L(T)$ are shown in Fig. 3e. The lattice component was obtained by subtracting the electronic contribution, estimated using the Wiedemann-Franz law with a commonly used approximation

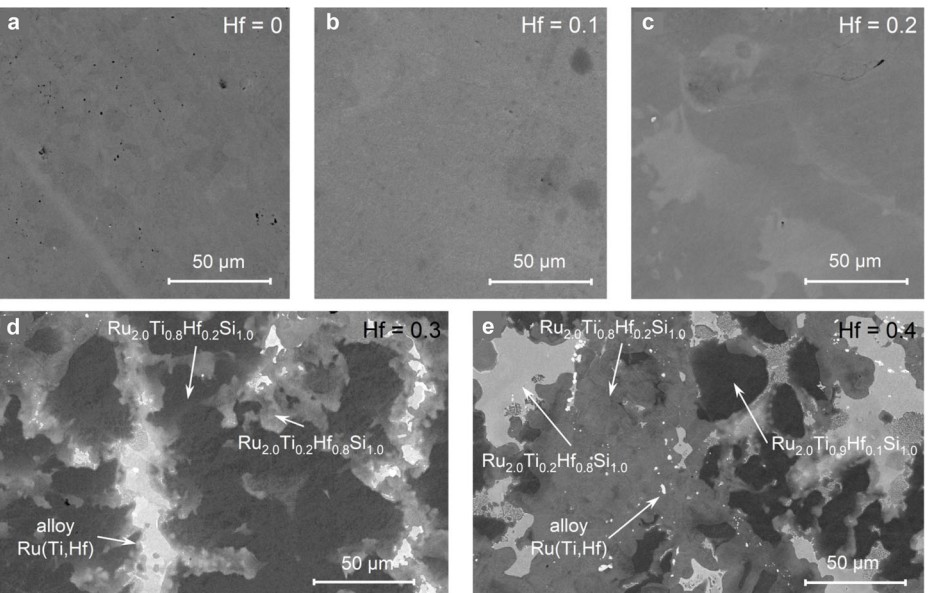

**Fig. 2 | Scanning electron microscopy micrographs of Ru₂Ti₁₋ₓHfₓSi acquired in back-scattered electron mode. a** to **c** show homogeneous microstructures without visible secondary phases for Hf concentrations $x \leq 0.2$. **d** and **e** show phase separation in samples with $x = 0.3$ and $0.4$, respectively. The compositions labeled in **d** and **e** for the full-Heusler phases and the Ru-(Ti,Hf) alloy phase (see white arrows) were determined by energy-dispersive x-ray spectroscopy.

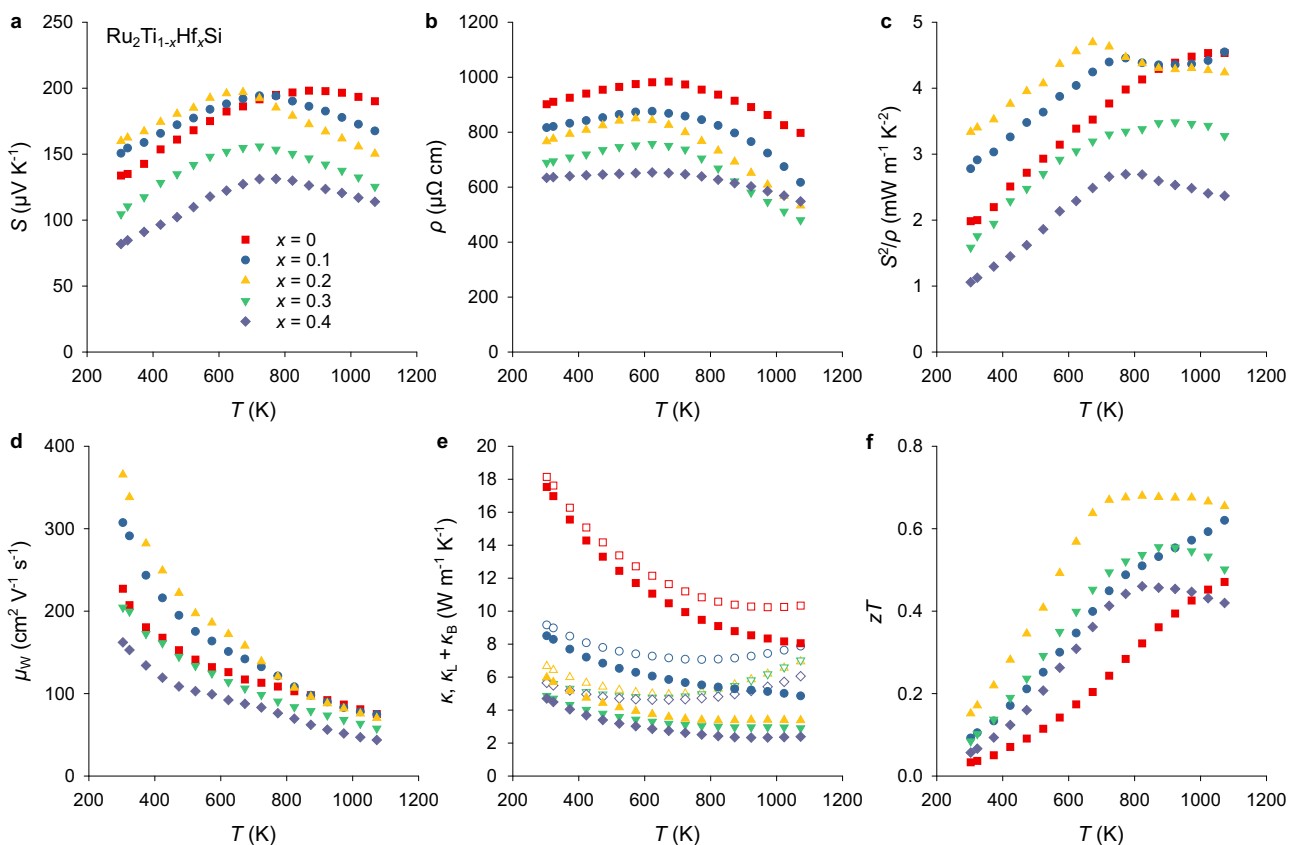

**Fig. 3 | Temperature-dependent thermoelectric properties of Ru₂Ti₁₋ₓHfₓSi.**
**a** Seebeck coefficient, **b** electrical resistivity, **c** power factor, **d** weighted mobility[68], **e** thermal conductivity and **f** dimensionless figure of merit. Open symbols in **e** denote the total thermal conductivity, whereas full symbols are the lattice plus bipolar contributions $\kappa_L + \kappa_B$.

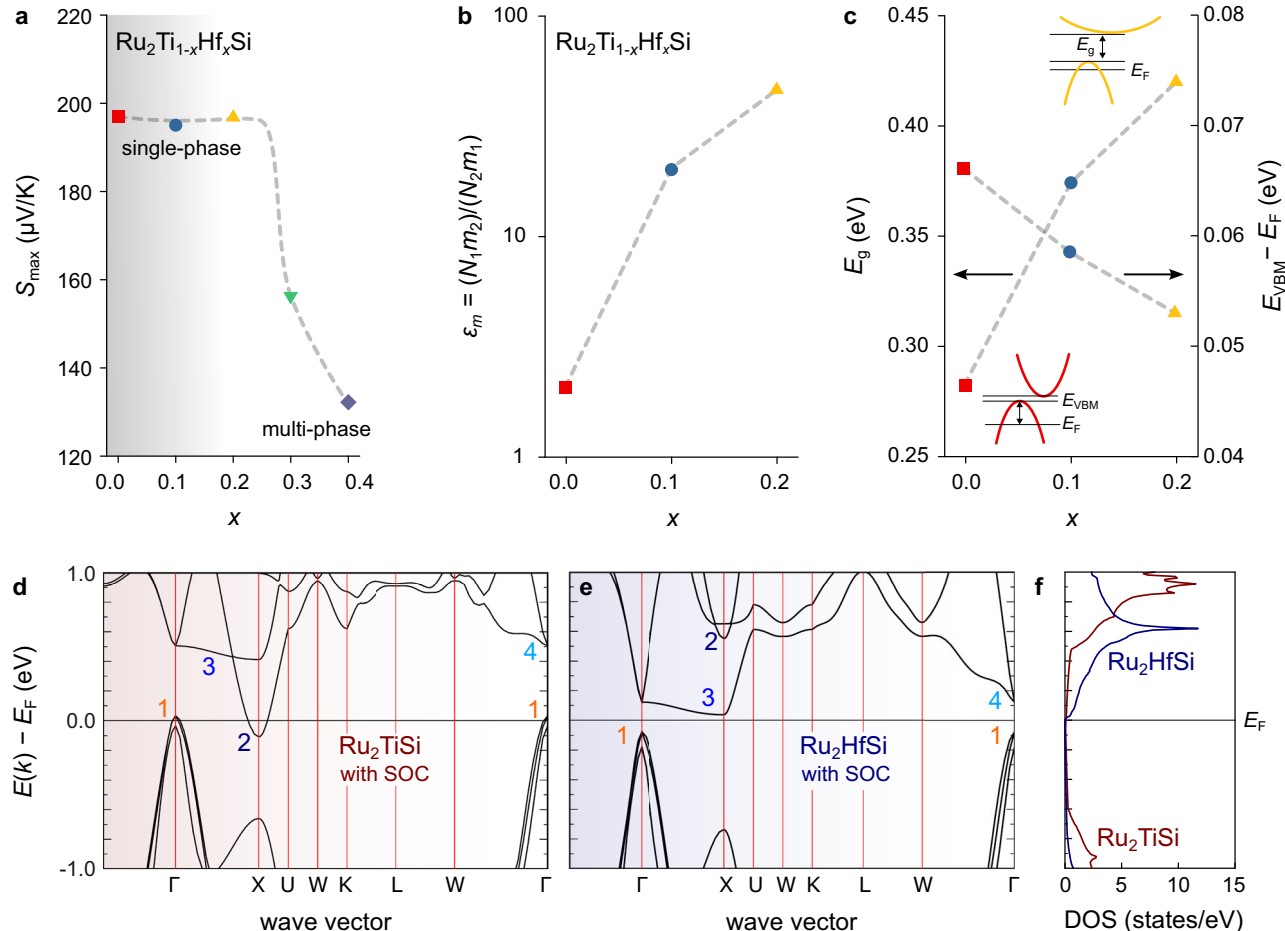

**Fig. 4 | Evolution of electronic structure with Hf substitution. a** Maximum Seebeck coefficient as a function of Hf concentration in $Ru_2Ti_{1-x}Hf_xSi$. The sudden drop around $x \sim 0.3$ coincides with the solubility limit. **b** Weighting parameter between conduction and valence band, extracted from least-squares fits of the temperature-dependent Seebeck coefficient. **c** Band gap and Fermi level position relative to the valence band edge, derived from least-squares fits of the temperature-dependent Seebeck coefficient employing a two-parabolic band model. Gray dashed lines in **a** to **c** are guides to the eye. **d** DFT band structure of $Ru_2TiSi$ and **e** $Ru_2HfSi$, calculated with spin−orbit coupling and using standard GGA-PBE exchange correlation functionals[79]. **f** Electronic densities of states corresponding to bandstructures in (**d**) and (**e**).

for the Lorenz number in TE semiconductors: $L = 1.5 + \exp(-|S|/116)$[69]. It should be noted that $\kappa_L(T)$ still includes the bipolar thermal conductivity contribution $\kappa_B$, which becomes only relevant at temperatures near the maximum of $S(T)$ and above.

Substitution of Ti with the much heavier and larger $5d$ element Hf introduces strong atomic mass and strain field fluctuations at the Y site, leading to enhanced phonon scattering in line with heavy-element substitution at the V site in $Fe_2VAl$-based fH systems[44,56,70]. As expected, this heavily impedes lattice-driven heat transport and significantly reduces $\kappa_L$ down to approximately 3.4 W m$^{-1}$ K$^{-1}$ in $Ru_2Ti_{0.8}Hf_{0.2}Si$ and down to around 2.3 W m$^{-1}$ K$^{-1}$ in $Ru_2Ti_{0.6}Hf_{0.4}Si$. The combination of this suppressed $\kappa_L$ and concurrently enhanced weighted mobility $\mu_W$ results in a maximum figure of merit of $zT_{max} \sim 0.7$ for $Ru_2Ti_{0.8}Hf_{0.2}Si$, sustained over a broad temperature range 700−1000 K. To the best of our knowledge, these values exceed those reported for any other fH bulk material to date.

To investigate changes in the electronic structure induced by Hf/Ti substitution, we employed a two-parabolic-band model to analyze the temperature- and doping-dependent evolution of $S(T)$ in $Ru_2Ti_{1-x}Hf_xSi$. For this purpose, we used the *SeeBand* code[71]−a recently developed fitting tool based on Boltzmann transport theory within a parabolic band framework−which enables an efficient analysis of temperature-dependent electronic transport properties.

## Electronic structure changes

Figure 4a shows the dependence of $S_{max}$ on $x$, while Fig. 4b, c present the electronic structure parameters obtained from fitting $S(T)$ for samples within the single-phase regime using a two-parabolic band model. The model includes three independent fitting parameters: (i) the position of the Fermi level relative to the valence band edge, $E_F$; (ii) the band gap between the valence band maximum and the CBM, $E_g$; and (iii) a weighting parameter $\epsilon_m = (N_1 m_2)/(N_2 m_1)$, which, alongside $E_F$, determines the relative contributions of the valence and conduction bands to the electrical conductivity. Here, $N_1$ and $N_2$ are the band degeneracies, and $m_1$ and $m_2$ are the effective masses of the valence and conduction bands, respectively.

The extracted fit parameters reveal consistent trends. Notably, $\epsilon_m$ increases sharply with $x$ by more than an order of magnitude. Additionally, the band gap $E_g$ increases with Hf content, while the Fermi level shifts closer to the valence band edge. These changes appear qualitatively consistent with expectations based on the electronic band structures and densities of states of the fully substituted Heusler compounds $Ru_2TiSi$ and $Ru_2HfSi$ from density functional theory (DFT) calculations, shown in Fig. 4d−f. Both Heusler compounds display a triply degenerate valence band maximum at Γ, which splits when considering spin orbit interactions, but different conduction band minima. For $Ru_2TiSi$, the conduction band minimum (CBM) is a dispersive Ti $e_g$ band at X, whereas for $Ru_2HfSi$ this band shifts upwards in

energy (likely due to the higher energy of Hf $5d$ states compared to Ti $3d$ states). At the same time, the flat Ru $e_g$ band along Γ−X gets pushed closer toward $E_F$ and becomes the new CBM. Also the conduction bands at X are pushed closer toward $E_F$, leading to an increase of the DOS effective mass at the conduction band side of the band gap (Fig. 4f).

We also note that, on the DFT level, the electronic structure of Ru$_2$TiSi is a semimetal with a slight overlap ( ≈ 0.12 eV) between the valence band at Γ and the conduction band at X, which is inconsistent with the large Seebeck effect and finite band gap $E_g ≈ 0.28$ eV, derived from our 2PB model. In Fe$_2$VAl, it was shown that an additional effective onsite Coulomb interaction $(U − J)$ for the $3d$ transition metal atoms is necessary to correctly describe the electronic structure and band gap[48,49]. This prompted us to investigate the effect of $(U − J)$, within the framework of DFT $+ (U − J)$[72], on the electronic structure of Ru$_2$TiSi. Results of these calculations are summarized in Figs. S3 and S4 and show that the band gap increases with $(U − J)$ and yields a semimetal−semiconductor crossover around $(U–J)_{Ti} = 0.6$ eV. For larger values, $(U–J)_{Ti} ≈ 1.5$ eV, we find optimal agreement with the experimental Seebeck coefficient and electronic transport calculations in the constant relaxation time approximation.

To estimate whether there is still room for improvement of the TE properties by optimizing the carrier concentration through co-substitution, we performed a more in-depth analysis of the best-performing sample Ru$_2$Ti$_{0.8}$Hf$_{0.2}$Si, which is close to but still well below the solubility limit of Hf. Figure S5 shows least-squares fits of the temperature-dependent electrical resistivity and Seebeck coefficient (black solid lines), which can be fitted simultaneously via an advanced self-consistent fitting algorithm leveraging the *SeeBand* code[71]. The fitting procedure minimizes the number of free parameters by fixing the electronic structure obtained from fitting $S(T)$ when modeling $ρ(T)$. A fit of $ρ(T)$ can then give information regarding the scattering times of the individual bands, which slightly modifies the theoretical $S(T)$ for the same electronic structure. A self-consistent iterative loop can derive the best solution for both measured transport properties. The framework for the fit algorithm is a two-parabolic band model with dominant acoustic phonon scattering. Future work should also take into consideration the role of optical phonons, which may be important in enabling intervalley scattering processes between the conduction bands at X and the valence bands at Γ.

Contrary to Fe$_2$VAl[73], there are no signatures of in-gap states and anomalous scattering off such localized impurity (antisite) states in Ru$_2$TiSi. The excellent agreement with experimental data−particularly above 400 K−shown in Fig. S5a, b highlight the robustness of the fit. Interestingly, slight deviations below 400 K, which should become even more pronounced for $T < 300$ K may indicate the relevance of a second valence band, which does not appear close to $E_F$ in DFT. For simplicity, however, a third band was not taken into consideration in our current analysis to reduce the number of free parameters and ambiguity of the derived electronic band structure model.

Figure S5c shows that by slightly adjusting the position of $E_F$, the power factor could be substantially improved up to around 7.5 mW m$^{-1}$ K$^{-2}$ at around 1100 K if $E_F$ could be lowered 70 meV deeper into the valence band. This could be achieved via Al co-substitution at the Si site and should increase $zT$ up to 0.8−0.9 assuming the same $κ_L$ as for Ru$_2$Ti$_{0.8}$Hf$_{0.2}$Si.

Figure 5 gives an overview of the best p- and n-type TE performances achieved so far in fH bulk materials. Ru$_2$Ti$_{0.8}$Hf$_{0.2}$Si from this work, reaching $zT_{max} = 0.7$, represents a record-high value among both p- and n-type fH materials studied up until now. We note that this value is in excellent agreement with earlier parabolic-band model predictions ($zT_{pred} \sim 0.76$) for the same carrier concentration and a heavy-element substitution of $x = 0.2$[63]. Since the solubility limit of Hf in Ru$_2$Ti$_{1−x}$Hf$_x$Si is limited to around $x = 0.26$, co-substitution with other heavy elements, such as Zr for Ti or Ge and Sn for Si should be explored

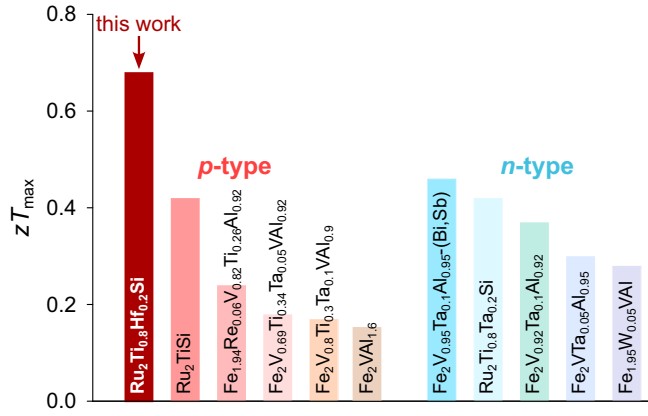

**Fig. 5 | Superior thermoelectric performance among full-Heuslers.** Comparison of maximum $zT$ for the best p- and n-type Fe$_2$VAl[50,53 - 55,59 - 61,81] and Ru$_2$TiSi-based[62] full-Heusler bulk thermoelectric materials. Ru$_2$Ti$_{0.8}$Hf$_{0.2}$Si from this work reaches the highest $zT$ achieved in full-Heusler systems up until now.

to reduce $κ_L$ even further. If this can be accomplished, there is a high likelihood that $zT > 1$ could be achieved in optimally doped Ru$_2$TiSi-based fHs.

In conclusion, we have investigated the fH compound series Ru$_2$Ti$_{1−x}$Hf$_x$Si and found a solubility limit of Hf and a miscibility gap between $0.26 ≤ x ≤ 0.8$ from both powder XRD and SEM. The TE properties of compositions $x = 0.1, 0.2, 0.3$, and $0.4$ were studied in a broad temperature range 300−1100 K. We found that within the single-phase regime, the maximum Seebeck coefficient remains almost the same but shifts towards lower temperatures. Due to the lack of Ti or Hf orbital contributions in the valence band electronic structure, the electrical resistivity is hardly affected by the substitution and disorder introduced thereby and does not increase as in the case of Ta substitution, where $E_F$ is shifted into the Ti/Ta $e_g$ conduction bands. Surprisingly, $ρ(T)$ even decreases with $x$ in Ru$_2$Ti$_{1−x}$Hf$_x$Si−an interesting subject for further investigation−leading to enhanced values of the power factor and weighted carrier mobility. Consequently, a relatively large $zT = 0.7$ could be achieved between 700−1000 K, which according to a two-parabolic band modeling analysis can be further improved by optimizing the carrier concentration through co-doping. Our results motivate further exploration of co-substituted and nanostructured Heusler compounds on the basis of Ru$_2$Ti$_{0.8}$Hf$_{0.2}$Si (e.g., with additional Ge/Si or Sn/Si alloying) and underscore the potential of screening novel semiconducting 24-valence-electron fH systems for TEs.

## Methods

### Synthesis

Samples of Ru$_2$Ti$_{1−x}$Hf$_x$Si ($x = 0, 0.1, 0.2, 0.3, 0.4$) were synthesized by arc melting stoichiometric amounts of high-purity elements: Ru rod (99.99 mass%), Ti ingot (99.99 mass%), Hf ingot (99.9 mass%, with 1.2 at.% of Zr), and Si shot (99.999 mass%), all supplied by Rare Metallic Co. (Japan) at National Institute for Materials Science (NIMS). To reduce the melting temperatures of Ru and Hf, we premelted them together with a stoichiometric amount of Ti. Subsequently, the required amount of Si was added and the mixture was arc melted once. Attempts to remelt the resulting ingot led to explosion, preventing further remelting. The arc-melted ingot was manually crushed inside an Ar-filled glovebox. The resulting powder was consolidated via spark plasma sintering (Dr.Sinter-1080, Fuji-SPS, Japan) in a diameter 10 mm graphite die under a uniaxial pressure of 50 MPa at 1773 K for 10 min in Ar atmosphere, with a heating rate of 100 K/min. All sintered samples were annealed at 1273 K under vacuum for three days, followed by

quenching. The second set of $Ru_2Ti_{1-x}Hf_xSi$ ($x$ = 0, 0.1, 0.2, 0.8, 1) were synthesized by high-frequency induction melting at TU Wien. Raw elements were of 99.99% purity for Ru, 99.95% for Ti, 99.99% for Hf and 99.9999% for Si.

## Characterization

Phase composition was analyzed by powder XRD using a Bragg-Brentano geometry in a $\theta$−$2\theta$ configuration (SmartLab, Rigaku Corporation, Japan). Scans were performed over a $2\theta$ range of 5° to 157° using monochromatic Cu$K\alpha_1$ radiation ($\lambda$ = 1.54056 Å), with a step size of 0.02° and a scanning speed of 1°/min. Crystal structure refinement was conducted using the WinCSD software package[74]. Microstructural features and elemental distributions were investigated via high-resolution SEM (HRSEM, SU8230, Hitachi, Japan) equipped with energy-dispersive X-ray spectroscopy (EDS, X-Max$^N$, Oxford Instruments, UK). The temperature-dependent electrical resistivity ($\rho$) and Seebeck coefficient ($S$) were simultaneously measured using the four-probe method on bar-shaped specimens (10 × 3 × 1.5 mm), oriented perpendicular to the SPS pressing direction. Measurements were carried out using a commercial ZEM-3 system (Advance-Riko, Japan). The total thermal conductivity ($\kappa$) was calculated as $\kappa = \chi \cdot C_p \cdot d$, where $\chi$ denotes thermal diffusivity measured by the laser flash technique (LFA 457 MicroFlash, Netzsch, Germany), $C_p$ is the specific heat capacity obtained via the comparative method using a pyroceram-9606 reference, and $d$ is the bulk density determined through the Archimedes method. To reduce radiative heat loss errors due to surface emissivity, the samples were coated with a thin graphite layer.

## Density functional theory calculations

Density functional theory (DFT) calculations were performed using the Vienna Ab Initio Simulation Package (VASP)[75,76]. Pseudo potentials for VASP were constructed according to the projector- augmented-wave (PAW) method as formulated in refs. [77,78]. Exchange correlations were treated in the generalized gradient approximation (GGA) as parametrized by Perdew, Burke and Ernzerhof (PBE)[79]. A high-precision plane wave energy cutoff of 400 eV was chosen in the calculations with integration over the first Brillouin zone (BZ) being performed using the tetrahedron method and approximately 4500 $k$ points in the irreducible part of the BZ. Relativistic effects were included by taking spin−orbit coupling into account in the Hamiltonian. DFT + ($U − J$) calculations were performed within the framework of ref. [72] and Boltzmann-transport calculations in the constant relaxation time approximation (CRTA) were done by using an adapted version of the BoltzTrap package[80].

# Data availability

The data that support the findings of this study are available in the main paper and its Supplementary Information.

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

## Acknowledgements

Financial support came from the Japan Science and Technology Agency (JST) project Mirai JPMJMI19A1. F.G. acknowledges a Director's Post-doctoral Fellowship through the Laboratory and Directed Research & Development (LDRD) program. A.P. acknowledges support from OeAD WTZ (Project HR 05/2024). The authors also acknowledge the TU Wien Bibliothek for financial support through its Open Access Funding Program. The authors would like to further acknowledge NIMS Nanofabrication Facility for SEM analysis and Material Analysis Station (NIMS, Japan) for the XRD analysis. DFT calculations were performed on the facilities of the Austrian Scientific Cluster (ASC).

## Author contributions

F.G. and I.S. synthesized samples and performed XRD experiments. I.S. investigated the microstructure and measured the temperature-dependent TE properties. R.P. performed DFT calculations. F.G. and M.P. conceptualized the study. F.G. and I.S. wrote the initial draft. A.P., T.M., and E.B. supervised the work and edited the manuscript.

## Competing interests

There are no competing interests.
