## [Transparent Peer Review file · Nature Communications]

Orbital-selective band engineering realizes high zT in p-type $\text{Ru}_2\text{Ti}_{1-x}\text{Hf}_x\text{Si}$ full-Heusler thermoelectrics

Corresponding Author: Dr Fabian Garmroudi

Version 0:

Reviewer comments:

Reviewer #1

(Remarks to the Author)

P-type $\text{Ru}_2\text{Ti}_{1-x}\text{Hf}_x\text{Si}$ full-Heusler bulk thermoelectrics with $zT = 0.7$

Authors have presented their work on the synthesis and study of a p-type thermoelectric material, a full-Heusler Ru_2TiSi doped with the isovalent atom Hf at the Ti site. Authors have studied these samples using X-ray diffraction, SEM-EDS, theoretical calculations, and thermoelectric properties. The work is detailed and shows a significant enhancement in the performance of the material under study.

Comment 1. Observing the XRD data presented carefully, it seems there is some broadening in the peak (220) from $x=0.2$ and further. Since broadening can be due to various reasons, is this due to the segregation of multiple phases of cubic $\text{Ru}_2(\text{Ti}/\text{Hf})\text{Si}$ -rich phases, stress due to doping with a heavier element, or something else? Kindly check the peak width (or FWHM) with the doping percentage.

Comment 2. The authors mentioned that there is no segregation for the $x=0.1$ sample in the BSE image, but different contrasts are clearly visible. Does the sample undergo a phase segregation between $x=0$ and $x=0.2$ sample which increases further with further doping of Hf at the Ti site? A sample with $x=0.5$ and $x=0.15$ can give a better perspective. On careful observation, it seems that the samples with $x>0.2$ have more than two contrasts visible in the BSE image. Please comment. This is also related to the peak broadening asked in comment 1.

Comment 3. The resistivity data show a maxima-like behavior between 600-700 K, after which resistivity decreases drastically, which roughly coincides with the S_{max} . Please explain.

Comment 4. The total thermal conductivity data shown in Fig. 3. shows increasing thermal conductivity for $x=0.2$ and beyond. What is the reason for this increase? Please explain.

Also, the slope of the lattice thermal conductivity has changed drastically for the samples after doping, showing a change in scattering mechanism.

Comment 5. From electronic structure calculation, it seems that Ru_2TiSi is a semimetal rather than a gapped material (Hf counterpart is shown as an indirect bandgap semiconductor). As far as I know, Ru_2TiSi has a bandgap of 0.2-0.3 eV (Theoretical). What is the bandgap of the undoped Ru_2TiSi sample authors have obtained? Please comment.

Comment 6. For proper understanding and completeness, since thermal conductivity decreases drastically, I would recommend including the expected theoretical density and experimental density should be added in the supplementary information.

The work is detailed and can be published after addressing the comments.

Reviewer #2

(Remarks to the Author)

In the present manuscript, the authors report their observation of synergetic reduction in lattice thermal conductivity and electrical resistivity in p-type full-Heusler Ru_2TiSi via tuning isovalent Hf doping at Ti sites. The material thus obtained has been found to exhibit a record-high figure of merit ($zT \approx 0.7$) between 700–1000 K, making it the most efficient full-Heusler bulk thermoelectric material reported to date. The authors were led to their experimental investigation following the observation of enhanced thermoelectric performance of Ru_2TiSi Heusler compounds with Ta substitution by Fujimoto et al. [Journal of Alloys and Compounds, vol. 969, p. 172345, Dec. 2023]. In a previous study, on the thermoelectric transport in Ru_2TiSi Full-Heusler Compounds [PRX Energy, vol. 4, no. 1, p. 013010, Mar. 2025], the present authors had identified p-type Ru_2TiSi as far more efficient than previously studied n-type compounds and demonstrated that this can be traced back

to much lighter and more mobile holes originating from dispersive valence bands. In our view, the present manuscript reports novel and impactful results and is acceptable for publication in the Nature Communications after revision. The following are our suggestions and comments to the authors:

Comment 1: The title of the manuscript should be improved to make the significance of the work more readily visible:

1. The title “P-type Ru₂Ti_{1-x}Hf_xSi full-Heusler bulk Thermoelectrics with zT=0.7” succinctly and accurately points to the main novel result at the center of the present research work. However, we believe that to make the significance of the result more readily visible, especially to the audience not specialized in Heusler-based TE materials, the title should place greater emphasis on the fact that, for the first time, competitive thermoelectric performance in half-Heusler TE materials was observed in a full-Heusler compound.

2. Also, in our view, the title may be made more interesting, and representative of the contents of the manuscript by mentioning that, the authors were prompted to work on synergistic tuning of lattice thermal conductivity and electronic transport properties of p-doped Ru₂TiSi by isovalent Hf-substitution at Ti sites from their scrutiny of prior experimental results on other related materials, their knowledge of physical considerations (electronic structure and orbital resolved character of band edges of Ru₂TiSi based materials, bonding in Heusler materials, Boltzmann transport, etc.) and, their ingenious two parabolic band and acoustic deformation potential theory model (2PB + Acoustic DPT model). This may help enhance the visibility of the work’s significance.

Comment 2: In addition to the references to the SeeBand code publication [npj Comput Mater 11, 171 (2025)], more details should be included in the main text on the specifics of the 2PB + Acoustic DPT model and the fitting procedure. Furthermore, the authors may also consider adding mathematical equations for the thermoelectric transport coefficients and the corresponding reduced expressions that arise in the 2PB + Acoustic DPT model.

Comment 3: From the authors’ textual description of the structure, it is indeed possible to discern the intended structure. However, for better readability by the audience, figures with structure models for the Ru₂TiSi should be presented, and the Hf substitution site pointed out in these figures.

Comment 4: Since, as opposed to high-throughput screening, the present work concerns unveiling transport in a specific novel material, it is highly suggested that in addition to complementing inference from experimental results with their ingenious and very relevant 2PB + Acoustic DPT model-based method, the authors also study the results derived from ab initio electronic structure calculations (and, if found possible and appropriate, maybe even consider including DFPT or more generalized DPT model calculations of charge carrier scattering rates) and compare them with the experiment + SeeBand inferences. The electronic structure calculations can be made more accurate by using hybrid functional-based methods or by employing DFT+U, as was also found to be necessary in Fe₂VAl [J. Mater. Chem. C, 2021,9, 2073-2085]. The impact of random doping on the electronic structure can be simulated in VASP using the virtual crystal approximation (VCA) routine. Here, it is conceivable that comparing and checking for consistency and deviations among experimental, experimental + SeeBand, and ab initio inferences can elucidate interesting physics and/or the strengths and limitations of the 2PB + Acoustic DPT model, as well as VCA for this material. The ab initio inferences may also be particularly relevant for exploring the curious reduction in electrical resistance with increasing doping. The manuscript is adequate for publication even without any of the additions mentioned in this comment; however, these additions are highly suggested.

Comment 5. In the theoretical treatment, a brief mention or explanation should be added on the neglect of electron-phonon scattering other than that involving long-wavelength longitudinal acoustic phonons, especially because the material’s Fermi surface has pockets at the Γ and X points of the Brillouin zone. For electron-phonon scattering between these pockets, the $q \rightarrow 0$ approximation (i.e., the perturbing potential due to the relevant phonon mode can be represented as a linear combination of the perturbing potentials due to uniform crystal deformations), used in standard deformation potential theory-based methods, is questionable.

Comment 6. Suggested improvements to ‘Supplementary Information, Section 1.3 Density functional theory calculations:

1. Manuscript Excerpt: “Exchange correlations were treated in the semi-local generalized gradient approximation (GGA) as parametrized by Perdew, Burke and Ernzerhof (PBE) [6].”

Suggestion/Comment: As is clear from Phys. Rev. Lett. 1996,77, 3865–3868, the Perdew, Burke, and Ernzerhof (PBE) parametrized generalized gradient approximation (GGA) is completely local electron density based and is not a ‘semi-local XC functional’.

2. Manuscript Excerpt: “Pseudopotentials were constructed according to the projector-augmented-wave (PAW) method.”

Suggestion/Comment: This statement should be rephrased in more accurate and unambiguous language. The current phrasing may lead to some degree of confusion among the audience.

Comment 7. We suggest the following minor changes in the introductory text:

1. Manuscript Excerpt: “Their chemical and electronic properties are governed by simple electron-counting rules, such as the Slater-Pauling principle, which enable the rational design of semiconducting ground states— particularly attractive for thermoelectric applications— by targeting an average valence electron count (VEC) of six valence electrons per atom, VEC = 6 [35–37]”

Suggestion/Comment: This portion of the text featuring a hyphenated and highly composite sentence be rephrased to improve clarity.

2. Manuscript Excerpt: “Notably, hH compounds with VEC = 6 tend to exhibit wider band gaps than their fH counterparts. “

Suggestion/Comment: A brief explanation should be added as to why hH compounds with VEC = 6 tend to exhibit wider band gaps than their fH counterparts.

3. Manuscript Excerpts: “As a result, hH systems are generally better suited for high-temperature thermoelectric applications [25, 38–41], since optimal thermoelectric performance is often achieved when the temperature reaches a fraction of the band gap $E_g \sim 10k_B T_{work}$, where k_B is the Boltzmann constant and T_{work} denotes the working temperature for optimal device operation [42].” AND “Initial investigations into the thermoelectric properties of n-type Ta-substituted $Ru_2Ti_{1-x}Ta_xSi$ systems revealed a $zT_{max} \sim 0.4$ at high temperatures, around 900K [62] due to a larger band gap compared to their Fe_2VAl -based relatives. “

Suggestion/Comment: It is advised that the underlying reasons why the optimal working temperature may be expected to be higher for thermoelectric materials with a higher band gap are mentioned, in addition to the cited reference. These points may be explored:

- a. The broadening of the Fermi-Dirac distribution function resulting from increased temperature shifts the Fermi level away from the respective band edges (to maintain charge neutrality), and how this leads to an increased $(E_{nk}-E_F)$ weighted moment of transport distribution function
- b. Bipolar effects at elevated temperatures are less significant in such materials because the Fermi window remains far enough away from the other band edge.

Comment 8: In the schematic insets in Fig. 4 (c), the authors may consider shifting the conduction band along the crystal momentum axis with respect to the valence band edge to correctly represent the indirect band gap as is suggested from DFT evidence. Admittedly, this risks some confusion, if appropriate captions are not added, since the indirect band gap signature was not inferred from the SeeBand fitting procedure. The authors are required to examine these aspects and determine whether making such changes is desirable, and if so, how to display the indirect band gap while mitigating any risk of aforementioned confusion.

Reviewer #3

(Remarks to the Author)

Comments to the authors (Manuscript id: NCOMMS-25-78915)

This manuscript reports a high thermoelectric figure of merit (zT) value of 0.7 in the temperature range 700 – 1000 K in the p-type $Ru_2Ti_{0.8}Hf_{0.2}Si$ full-Heusler compound. This is notably a large value reported in any n-type or p-type full-Heusler compound. This work opens the possibility of developing high performance full-Heusler thermoelectric materials for mid-to-high temperature range applications. Therefore, I recommend this manuscript to be considered for publication in Nature Communication after minor revision.

1. The significantly high $zT \sim 0.7$ in the Ru_2TiSi system compared to Fe_2VAl (0.3-0.4) full-Heusler system is interesting. How feasible would it be in terms of societal applications considering the cost and abundance of Ru vs Fe?
2. The authors claim samples $x \leq 0.2$ to be single phase from both XRD and SEM-EDS measurements. There is a minor peak at 40° begins appearing when Hf-content $x > 0.1$, Fig. 1(a). Also, in Figs. 2 (a-c), there are different contrast regions visible. It would be better if authors could provide EDS measurements of these samples at different regions, along with elemental mapping.
3. Fig. 3(a); Hf-doped ($x \leq 0.2$) samples show profound bipolar behavior compared to the non-doped. Why? Please include explanation in the manuscript.
4. Fig. 4(c); the increment in the band gap, E_g is higher an order than that of the reduction in the $E_{VBM}-E_{CF}$ in the Hf-doped samples. However, higher bipolar behaviour can be noticed in the Hf-doped than non-doped. Why? The explanation should be added in the manuscript.
5. Did the authors measure Hall carrier concentration and mobility? It would be better to include this data in the manuscript to better explain the transport properties.
6. The samples were synthesized at two different laboratories NIMS and TU Wien. It is difficult to understand whether the presented thermoelectric properties belong to the samples synthesized at NIMS or at TU Wien. Also, it would be better if the comparison of the thermoelectric properties of the similar compositions synthesized at the different laboratories will be included in the supporting information, because the followed synthesis route at both the laboratories is different from each other.
7. Page #2; paragraph, “To investigate the...by XRD (Fig. 2c, d).”; last sentence, “In contrast...by XRD (Fig. 2c, d).” :- please check and correct the figure number.

Reviewer #4

(Remarks to the Author)

Version 1:

Reviewer comments:

Reviewer #1

(Remarks to the Author)

The authors have satisfactorily addressed all the previous comments and changes were made in the manuscript accordingly. I believe the manuscript can be published in the current form.

Reviewer #2

(Remarks to the Author)

The authors have answered all the queries raised by us during the review. The revised manuscript may accepted for publication.

Reviewer #3

(Remarks to the Author)

Comments to the authors (Manuscript id: NCOMMS-25-78915A)

The authors have satisfactorily addressed all my comments. They have also included the results of the suggested EDX and Hall measurements. The manuscript now provides a thorough and well supported study of the microstructure and thermoelectric properties of the Ru₂Ti_{1-x}Hf_xSi full-Heusler compounds. Therefore, I recommend this manuscript for publication in Nature Communications.

Reviewer #4

(Remarks to the Author)

Reviewer Comments, Author Responses, and Manuscript Changes

1 Reviewer #1

Authors have presented their work on the synthesis and study of a p -type thermoelectric material, a full-Heusler Ru_2TiSi doped with the isovalent atom Hf at the Ti site. Authors have studied these samples using X-ray diffraction, SEM-EDS, theoretical calculations, and thermoelectric properties. The work is detailed and shows a significant enhancement in the performance of the material under study.

Author response: We thank the reviewer for their positive feedback on our manuscript. Below, we provide point-by-point replies to the reviewer's comments.

1. Observing the XRD data presented carefully, it seems there is some broadening in the peak (220) from $x = 0.2$ and further. Since broadening can be due to various reasons, is this due to the segregation of multiple phases of cubic $\text{Ru}_2(\text{Ti}/\text{Hf})\text{Si}$ -rich phases, stress due to doping with a heavier element, or something else? Kindly check the peak width (or FWHM) with the doping percentage.

Author response: Thank you for pointing this out. A noticeable broadening of the (220) reflection is observed in Hf-doped Ru_2TiSi samples, whereas no peak splitting occurs. As suggested, we calculated the FWHM for all compositions (Table 1) and confirmed a progressive broadening of the (220) peak up to $x = 0.3$. This broadening most likely originates from minor but increasingly pronounced Hf compositional variations within the Ru_2TiSi phase, consistent with the BSE images and EDX spectroscopy (see response to the next comment). At higher Hf contents ($x \geq 0.3$), the broadening saturates, which can be attributed to reaching the solubility limit of Hf at the Ti site.

Table 1: Crystal structure refinement results of $\text{Ru}_2\text{Ti}_{1-x}\text{Hf}_x\text{Si}$ full-Heusler alloys obtained at NIMS, including lattice parameter a , isotropic atomic displacement parameters B_{ISO} , site occupancy factor (SOF), phase fractions (wt.%), weighted profile R -factor (R_{wp}), profile R -factor (R_p), FWHM for (220) peak, calculated and experimental density (ρ_{calc} , ρ_{exp})

Sample	Ru_2TiSi	$\text{Ru}_2\text{Ti}_{0.9}\text{Hf}_{0.1}\text{Si}$	$\text{Ru}_2\text{Ti}_{0.8}\text{Hf}_{0.2}\text{Si}$	$\text{Ru}_2\text{Ti}_{0.7}\text{Hf}_{0.3}\text{Si}$	$\text{Ru}_2\text{Ti}_{0.6}\text{Hf}_{0.4}\text{Si}$
Diffractometer	SmartLab				
Space group	$Fm\bar{3}m$				
2θ and θ/γ (max)	157 / 0.636				
a , Å	5.97624(3)	5.99394(3)	6.00933(9)	6.0250(1)	6.02436(8)
$3*B_{\text{ISO}}$					
4a (Ti)	0.50(2)	0.68(13)	0.44(5)	0.18(3)	0.19(10)
4b (Si)	1.1(1)	0.95(9)	0.93(8)	0.72(6)	0.9(4)
4c (Ru)	0.79(4)	0.42(5)	0.33(3)	0.25(2)	0.35(7)
$3*\text{SOF}$					
4a (Ti-Hf)	1	0.91(2)-0.09(2)	0.81(2)-0.19(2)	0.76(2)-0.24(2)	0.74(2)-0.26(2)
4b (Si)	1	1	1	1	1
4c (Ru)	0.98(1)	0.98(1)	1	0.99(3)	1
Main phase, wt.%	100	100	100	94.7(8)	91.9(7)
Secondary phase, wt.%	0	0	0	5.3(8)	8.1(7)
R_{wp} , %	5.2	6.9	8.2	9.4	10.8
R_p , %	4.9	6.5	5.8	9.2	11.5
FWHM	0.188(2)	0.256(2)	0.289(2)	0.350(3)	0.349(3)
ρ_{calc} , g/cm ³	8.65	8.98	9.31	9.64	10.04
ρ_{exp} , g/cm ³	8.17	8.76	9.04	9.41	9.8

2. The authors mentioned that there is no segregation for the $x = 0.1$ sample in the BSE image, but different contrasts are clearly visible. Does the sample undergo a phase segregation between $x=0$ and $x = 0.2$ sample which increases further with further doping of Hf at the Ti site? A sample with $x = 0.5$ and $x = 0.15$ can give a better perspective. On careful observation, it seems that the samples with $x > 0.2$ have more than two contrasts visible in the BSE image. Please comment. This is also related to the peak broadening asked in comment 1.

Author response: For this work, we used a photodiode backscattered electron detector (PDBSE), which, compared to conventional BSE detectors, provides higher sensitivity to compositional contrast and reduced sensitivity to surface topography (e.g., polishing quality) Goldstein et al. (2017); Gernert and Berger (2016). This allows the visualization of even very minor compositional variations, as observed for the samples with $x = 0.1$ and $x = 0.2$. The EDX maps acquired from the same regions (see Fig. R1) exhibit a homogeneous elemental distribution for all elements except Hf, indicating that the compositional variations detected in the BSE images for $x = 0.1$ and $x = 0.2$ are minor. Because Hf has a much larger atomic number (178.485 a.m.u.) compared to the lighter elements in the alloy (e.g., Si with 28.085 a.m.u.), even small fluctuations in local Hf concentration can lead to noticeable changes in PDBSE contrast. For instance, in $\text{Ru}_2\text{Ti}_{0.9}\text{Hf}_{0.1}\text{Si}$, EDX mapping reveals regions with slightly reduced Hf content, and EDX point analysis confirms that these variations range from 2.0 at.% to 3.1 at.%, which is not significant and remains consistent with a single-phase material. In samples with $x \geq 0.3$, the situation is different. Here, phase segregation takes place, and both EDX maps and point analysis clearly show distinct regions with different elemental concentrations. The

contrast variations observed in the BSE images for these compositions, therefore, originate from the formation of compositionally different alloys, where deviations are large enough to produce strong atomic-number contrast.

Figure R1: SEM micrographs of $\text{Ru}_2\text{Ti}_{1-x}\text{Hf}_x\text{Si}$ with $x = 0, 0.1, 0.2, 0.3, \text{ and } 0.4$, alongside with EDX point spectra and element mapping.

3. The resistivity data show a maxima-like behavior between 600-700 K, after which resistivity decreases drastically, which roughly coincides with the S_{max} . Please explain.

Author response: The resistivity maximum, which coincides roughly with the Seebeck maximum arises due to bipolar conduction, i.e. charge carriers becoming thermally excited from the conduction band across the band gap. In nominally undoped Ru_2TiSi and Hf-substituted compositions, the Fermi level is situated very close to the band edge, which results in bipolar conduction at high temperatures. This behavior is fully reproduced by our two-parabolic band model and is typical for narrow-gap semiconductors, where the Fermi level lies close to the band edges. In Fe_2VAI -based systems, the maximum occurs at lower temperatures due to the narrower band gap.

4. The total thermal conductivity data shown in Fig. 3. shows increasing thermal conductivity for $x = 0.2$ and beyond. What is the reason for this increase? Please explain.

Also, the slope of the lattice thermal conductivity has changed drastically for the samples after doping, showing a change in scattering mechanism.

Author response: This behavior is typical of narrow-gap semiconductors exhibiting bipolar charge transport. The increase of the "lattice" thermal conductivity is actually a bipolar contribution, which becomes important around the same temperature where the Seebeck coefficient displays its maximum. We simply subtracted the electronic part via the Wiedemann-Franz law, which does not include the bipolar term. Thus, the bipolar term still shows up in the lattice part. For clarification, we specified that κ_{lat} is actually $\kappa_{\text{lat}} + \kappa_{\text{bip}}$.

Also, in stoichiometric Ru_2TiSi , the dominant scattering processes are phonon-phonon Umklapp scattering processes at high temperatures as would be expected. Upon Hf substitution, which creates strong strain and mass fluctuations since it is much heavier than Ti, point defect scattering becomes important. This is fully consistent with other heavy element-substitution studies in full-Heusler compounds, such as $\text{Fe}_2\text{V}_{1-x}\text{W}_x\text{Al}$ Hinterleitner et al. (2020). We provided these explanations in the updated manuscript.

5. From electronic structure calculation, it seems that Ru_2TiSi is a semimetal rather than a gapped material (Hf counterpart is shown as an indirect bandgap semiconductor). As far as I know, Ru_2TiSi has a bandgap of 0.2-0.3 eV (Theoretical). What is the bandgap of the undoped Ru_2TiSi sample authors have obtained? Please comment.

Author response: We thank the referee for pointing this out. This is indeed interesting. Indeed, we obtain from our parabolic band model, a band gap of 0.2-0.3 eV. DFT calculations of transition-metal-based systems are known to underestimate band gaps due to the importance of electronic correlation effects. For the sister compound Fe_2VAl , one also obtains a semimetallic band structure with large overlap of the valence and conduction bands when using standard GGA-PBE exchange correlation functionals in the electronic structure DFT calculations. However, it can be shown that employing an additional effective on-site Coulomb repulsion $U - J$ for the Fe or V- d states can increase the band gap and recover consistent values with experimental results Anand et al. (2020), Hinterleitner et al. (2021). Motivated by the reviewer's comment, we have performed additional DFT+($U - J$) calculations to study the effect of on-site Coulomb repulsion among the transition metal electrons on the band gap. Indeed, one also finds a band gap increase in Ru_2TiSi with increasing $U - J$ for the Ti $3d$ states (see Fig. R2 below). Around $(U - J)_{\text{Ti}} = 0.6$ eV, a semimetal-semiconductor transition takes place and for $(U - J)_{\text{Ti}} \approx 1.5$ eV, we find optimal agreement with the experimental Seebeck coefficients (see Fig. R3 below). We have added a small discussion of this in the main text and added Figs. R2 and R3 to the Supplementary Information.

6. For proper understanding and completeness, since thermal conductivity decreases drastically, I would recommend including the expected theoretical density and experimental density should be added in the supplementary information.

Author response: Thank you for the suggestion. We have included both theoretical and experimental densities in Table 1 and in the Supplementary Information.

2 Reviewer #2

In the present manuscript, the authors report their observation of synergetic reduction in lattice thermal conductivity and electrical resistivity in p -type full-Heusler Ru_2TiSi via tuning isovalent Hf doping at Ti sites. The material thus obtained has been found to exhibit a record-high figure of merit ($zT \approx 0.7$) between 700–1000 K, making it the most efficient full-Heusler bulk thermoelectric material reported to date. The authors were led to their experimental investigation following the observation of enhanced thermoelectric performance of Ru_2TiSi Heusler compounds with Ta substitution by Fujimoto et al. [Journal of Alloys and Compounds, vol. 969, p. 172345, Dec. 2023]. In a previous study, on the thermoelectric transport in Ru_2TiSi Full-Heusler Compounds [PRX Energy, vol. 4, no. 1, p. 013010, Mar. 2025], the present authors had identified p -type Ru_2TiSi as far more efficient than previously studied n -type compounds and demonstrated that this can be traced back to much lighter and more mobile holes originating from dispersive valence bands. In our view, the present manuscript reports novel and impactful results and is acceptable for publication in the Nature Communications after revision. The following are our suggestions and comments to the authors:

Author response: We genuinely appreciate the reviewers' valuable time, comments and suggestions how to further improve the impact of our work. We have carefully gone through all the comments and made corresponding changes in the manuscript. Please see below, in blue, our point-by-point replies to the reviewers' comments and suggestions.

1. The title of the manuscript should be improved to make the significance of the work more readily visible:

a. The title " P -type $\text{Ru}_2\text{Ti}_{1-x}\text{Hf}_x\text{Si}$ full-Heusler bulk Thermoelectrics with $zT = 0.7$ " succinctly and accurately points to the main novel result at the center of the present research work. However, we believe that to make the significance of the result more readily visible, especially to the audience not specialized in Heusler-based TE materials, the title should place greater emphasis on the fact that, for the first time, competitive thermoelectric performance in half-Heusler TE materials was observed in a full-Heusler compound.

b. Also, in our view, the title may be made more interesting, and representative of the contents of the manuscript by mentioning that, the authors were prompted to work on synergistic tuning of lattice thermal conductivity and electronic transport properties of p -doped Ru_2TiSi by isovalent Hf-substitution at Ti sites from their scrutiny of prior experimental results on other related materials, their knowledge of physical considerations (electronic structure and orbital resolved character of band edges of Ru_2TiSi based materials, bonding in Heusler materials, Boltzmann transport, etc.) and, their ingenious two parabolic band and acoustic deformation potential theory model (2PB + Acoustic DPT model). This may help enhance the visibility of the work's significance.

Author response: Thank you for sharing your opinion and giving some suggestions on how to improve the title. Initially, we chose a simple, short title that focuses on the main result—the enhanced zT . However, we agree with the reviewers that putting some emphasis on the mechanism and orbital-guided and selective substitution could make the title sound even more interesting.

Thus, we rephrased the title to: "Orbital-selective band engineering realizes $zT = 0.7$ in p -type $\text{Ru}_2\text{Ti}_{1-x}\text{Hf}_x\text{Si}$ full-Heusler thermoelectrics"

2. In addition to the references to the SeeBand code publication [npj Comput Mater 11, 171 (2025)], more details should be included in the main text on the specifics of the 2PB + Acoustic DPT model and the fitting procedure. Furthermore, the authors may also consider adding mathematical equations for the thermoelectric transport coefficients and the corresponding reduced expressions that arise in the 2PB + Acoustic DPT model.

Author response: We thank the reviewers for their suggestion but we do not think that it is necessary to provide more details in the paper, since we clearly added a reference to the SeeBand software paper [npj Comput Mater 11, 171 (2025)], where all the details of the modeling framework, including all relevant equations as well as a video guide on how to use it are presented. This allows in principle anybody to reproduce and model the same data without requiring them to write their own code based on the same theoretical framework. Thus, we don't think that it is necessary to write down virtually the same equations and description in this paper again. Also, we explained how the model works and which parameters are used in the fitting procedure, which is the most interesting part to an unfamiliar reader anyways.

3. From the authors' textual description of the structure, it is indeed possible to discern the intended structure. However, for better readability by the audience, figures with structure models for the Ru_2TiSi should be presented, and the Hf substitution site pointed out in these figures.

Author response: Thank you for the suggestion and this is a good idea indeed. We have included the crystal structure in Fig.1 and pointed out the Ti/Hf positions.

4. Since, as opposed to high-throughput screening, the present work concerns unveiling transport in a specific novel material, it is highly suggested that in addition to complementing inference from experimental results with their ingenious and very relevant 2PB + Acoustic DPT model-based method, the authors also study the results derived from ab initio electronic structure calculations (and, if found possible and appropriate, maybe even consider including DFPT or more generalized DPT model calculations of charge carrier scattering rates) and compare them with the

experiment + SeeBand inferences. The electronic structure calculations can be made more accurate by using hybrid functional-based methods or by employing DFT+U, as was also found to be necessary in Fe₂VAI [J. Mater. Chem. C, 2021,9, 2073-2085]. The impact of random doping on the electronic structure can be simulated in VASP using the virtual crystal approximation (VCA) routine. Here, it is conceivable that comparing and checking for consistency and deviations among experimental, experimental + SeeBand, and ab initio inferences can elucidate interesting physics and/or the strengths and limitations of the 2PB + Acoustic DPT model, as well as VCA for this material. The ab initio inferences may also be particularly relevant for exploring the curious reduction in electrical resistance with increasing doping. The manuscript is adequate for publication even without any of the additions mentioned in this comment; however, these additions are highly suggested.

Author response: We thank the reviewer for these suggestions yet also acknowledging the relevance of our 2PB model. Indeed, more sophisticated first-principles calculations of the electron-phonon scattering rates and the transport properties would be desirable and certainly enhance our manuscript. However, given the present authors, we do not have the capability to perform such transport calculations and would require additional external collaborations. We think that full ab initio transport calculations of electron-phonon interactions and scattering rates would probably deserve their own paper as well.

That being said, we have performed additional DFT+($U - J$) calculations to see how the electronic structure changes upon employing an additional effective on-site Coulomb repulsion, which resulted in better agreement with the transport properties of Fe₂VAI as pointed out by the referees. We also performed simple CRTA calculations (which should at least describe the Seebeck coefficient reasonably well) and compared them to our experimental data. We find that the band gap increases when taking into account an additional on-site Coulomb interaction for the Ti d states (see Fig. R2 below), which is in better agreement with the large Seebeck coefficients and the enhanced band gap derived from our 2PB model. Around $(U - J)_{\text{Ti}} \approx 1.5$ eV, we find optimal agreement with the experimental Seebeck coefficient (see Fig. R3). We have added Fig. R2 and R3 to our Supplementary Information.

Figure R2: **a**, Band gap as a function of the effective on-site Coulomb interaction $(U - J)_{\text{Ti}}$ for the Ti $3d$ states in Ru₂TiSi from DFT+($U - J$) calculations. The band gap increases as a function of $(U - J)_{\text{Ti}}$ and a semimetal-semiconductor crossover occurs at $(U - J)_{\text{Ti}} \approx 0.6$ eV. Values of around $(U - J)_{\text{Ti}} \approx 1.5$ eV produce the best agreement with the experimental Seebeck coefficient and transport-derived band gap. **b**, Density of states from DFT+($U - J$) calculations corresponding to the recommended value $(U - J)_{\text{Ti}} \approx 1.5$ eV.

5. In the theoretical treatment, a brief mention or explanation should be added on the neglect of electron-phonon scattering other than that involving long-wavelength longitudinal acoustic phonons, especially because the material's Fermi surface has pockets at the Γ and X points of the Brillouin zone. For electron-phonon scattering between these pockets, the $q \rightarrow 0$ approximation (i.e., the perturbing potential due to the relevant phonon mode can be represented as a linear combination of the perturbing potentials due to uniform crystal deformations), used in standard deformation potential theory-based methods, is questionable.

Author response: We appreciate the referee's insightful remark regarding the neglect of electron-phonon scattering channels beyond long-wavelength longitudinal acoustic phonons, particularly in light of the Fermi-surface pockets located at both the Γ and X points. We agree that, for intervalley processes connecting these pockets, the $q \rightarrow 0$ approximation inherent to deformation-potential-based treatments is not strictly justified, and that a full first-principles evaluation of the momentum-dependent electron-phonon matrix elements would be required to capture such processes accurately.

At present, however, we do not have the computational capability to perform full ab initio electron-phonon calculations (including mode- and momentum-resolved coupling strengths and lifetimes). For this reason, our theoretical analysis relies on simplified models, such as the two-parabolic-band (2PB) framework with acoustic-phonon deformation-potential scattering. Despite its simplicity, this approach has previously been shown to reproduce quite

accurately the Seebeck coefficients of various full-Heusler systems producing reliable band-gap estimates that are in agreement with DFT and other experimental probes Garmroudi et al. (2025).

Moreover, our new constant-relaxation-time approximation (CRTA) calculations—despite fully neglecting electron-phonon scattering—also reproduce the temperature dependence and magnitude of the Seebeck coefficient remarkably well (see Fig.R3 below). The fact that two conceptually different, simplified frameworks achieve similarly good agreement with the experimental thermopower strongly suggests that the Seebeck effect in these materials is not highly sensitive to the detailed momentum structure of the electron-phonon coupling but likely depends mostly on the group velocities and the gap.

We fully agree with the referee, however, that the resistivity could depend more sensitively on intervalley electron-phonon processes and on the detailed structure of scattering matrix elements beyond the long-wavelength acoustic limit. A quantitative description of the resistivity therefore requires a more complete treatment of electron-phonon interactions than is currently not feasible for us. We consider this an important and worthwhile direction for future work.

Figure R3: **a**, Temperature-dependent Seebeck coefficient of undoped Ru_2TiSi and comparison to theoretical calculations. A least-squares fit employing a two-parabolic band model can describe the experimental data very well. Transport calculations within the constant relaxation time approximation derived from the DFT band structure can only describe the experimental data when an additional effective on-site Coulomb interaction $(U - J)_{\text{Ti}} \approx 1.5 \text{ eV}$ for the Ti $3d$ states is taken into consideration. **b**, Temperature-dependent electrical resistivity. The constant relaxation time was chosen such that the calculated resistivity is normalized to the maximum of the experimental $\rho(T)$ curve.

6. Suggested improvements to ‘Supplementary Information, Section 1.3 Density functional theory calculations:

1. Manuscript Excerpt: “Exchange correlations were treated in the semi-local generalized gradient approximation (GGA) as parametrized by Perdew, Burke and Ernzerhof (PBE) [6].” Suggestion/Comment: As is clear from Phys. Rev. Lett. 1996,77, 3865–3868, the Perdew, Burke, and Ernzerhof (PBE) parametrized generalized gradient approximation (GGA) is completely local electron density based and is not a ‘semi-local XC functional’.

Author response: The referee’s remark is not fully correct. The approach of Perdew et al. goes beyond what is commonly understood as “local density approximation” which has its own meaning. There exists a variety of “local density approximations” (LDA) with specific parametrizations of the exchange-correlation functional. The “gradient approximation” (GGA in our case) also includes the gradient of the density in a local form. If you want to be precise you could call GGA a “local density and local density gradient approximation”. Because the gradient of a function gives some information about the change of the function at a point – i.e. its immediate neighbourhood going beyond strict locality – one might describe the gradient functional as a semi-local approach. The whole discussion has anyway no importance whatsoever for the paper. Therefore, we remove semi-local and write: “... were treated in the generalized gradient approximation ...”

2. Manuscript Excerpt: “Pseudopotentials were constructed according to the projector-augmented-wave (PAW) method.” Suggestion/Comment: This statement should be rephrased in more accurate and unambiguous language. The current phrasing may lead to some degree of confusion among the audience.

Author response: We rephrase: “Pseudo potentials for VASP were constructed according to the projector-augmented-wave (PAW) method as formulated in Refs. [4, 5].”

7. We suggest the following minor changes in the introductory text:

1. Manuscript Excerpt: “Their chemical and electronic properties are governed by simple electron-counting rules, such as the Slater-Pauling principle, which enable the rational design of semiconducting ground states– particularly attractive for thermoelectric applications– by targeting an average valence electron count (VEC) of six valence electrons

per atom, VEC = 6 [35–37]”. Suggestion/Comment: This portion of the text featuring a hyphenated and highly composite sentence be rephrased to improve clarity.

Author response: Thanks, we have split this into two sentences and rephrased to: ”Their chemical and electronic properties follow simple electron-counting rules such as the Slater–Pauling principle. This enables the rational design of semiconducting ground states—particularly relevant for thermoelectric applications—by targeting an average valence electron count of six per atom (VEC = 6) [35–37].”

2. Manuscript Excerpt: “Notably, hH compounds with VEC = 6 tend to exhibit wider band gaps than their fH counterparts.” Suggestion/Comment: A brief explanation should be added as to why hH compounds with VEC = 6 tend to exhibit wider band gaps than their fH counterparts.

Author response: Thanks, we extended this sentence to: ”Notably, hH compounds with VEC = 6 tend to exhibit wider band gaps than their fH counterparts, which can be attributed to the reduced symmetry and the reduced d – d hybridization. In fH compounds, the additional X atom introduces strong d – d hybridization that broadens the valence and conduction manifolds, often narrowing or even closing the band gap.”

3. Manuscript Excerpts: “As a result, hH systems are generally better suited for high-temperature thermoelectric applications [25, 38–41], since optimal thermoelectric performance is often achieved when the temperature reaches a fraction of the band gap $E_g \sim 10 k_B T_{\text{work}}$, where k_B is the Boltzmann constant and T_{work} denotes the working temperature for optimal device operation [42].” AND “Initial investigations into the thermoelectric properties of n -type Ta-substituted $\text{Ru}_2\text{Ti}_{1-x}\text{Ta}_x\text{Si}$ systems revealed a $zT_{\text{max}} \sim 0.4$ at high temperatures, around 900K [62] due to a larger band gap compared to their Fe_2VAl -based relatives.” Suggestion/Comment: It is advised that the underlying reasons why the optimal working temperature may be expected to be higher for thermoelectric materials with a higher band gap are mentioned, in addition to the cited reference. These points may be explored:

a. The broadening of the Fermi-Dirac distribution function resulting from increased temperature shifts the Fermi level away from the respective band edges (to maintain charge neutrality), and how this leads to an increased ($E_{nk}-E_F$) weighted moment of transport distribution function

b. Bipolar effects at elevated temperatures are less significant in such materials because the Fermi window remains far enough away from the other band edge.

Author response: Thank you for these suggestions. It is true that this is an inter-disciplinary journal but we still expect the majority of the readers will be from the TE community. We think that these are well enough established concepts that most people in the thermoelectric community should understand them. Optimal performance is usually obtained next to the bipolar regime since bipolar conduction reduces the Seebeck effect (and power factor) and also increases the thermal conductivity. Thus, a semiconducting material’s performance degrades when the bipolar regime is reached. This is true for almost all thermoelectric semiconductors. In some thermoelectrics with wider band gaps, the band gap may be so wide that transport always remains within the unipolar regime. However, wider band gap materials (such as oxides) are often characterized by less dispersive bands and lower valley degeneracy, which is why Bi-based topological insulators with narrow band gaps, for instance, have the highest room-temperature performance.

8. In the schematic insets in Fig. 4 (c), the authors may consider shifting the conduction band along the crystal momentum axis with respect to the valence band edge to correctly represent the indirect band gap as is suggested from DFT evidence. Admittedly, this risks some confusion, if appropriate captions are not added, since the indirect band gap signature was not inferred from the SeeBand fitting procedure. The authors are required to examine these aspects and determine whether making such changes is desirable, and if so, how to display the indirect band gap while mitigating any risk of aforementioned confusion.

Author response: Thank you for the suggestion. We like the idea and shifted it to match the DFT picture. However, we did not label or draw an x-axis since there occurs no momentum dependence in our fitting routine.

3 Reviewer #3

This manuscript reports a high thermoelectric figure of merit (zT) value of 0.7 in the temperature range 700–1000 K in the p -type $\text{Ru}_2\text{Ti}_{0.8}\text{Hf}_{0.2}\text{Si}$ full-Heusler compound. This is notably a large value reported for any n -type or p -type full-Heusler compound. This work opens the possibility of developing high-performance full-Heusler thermoelectric materials for mid- to high-temperature applications. Therefore, I recommend that this manuscript be considered for publication in *Nature Communications* after minor revision.

Author response: First and foremost, we are very happy to hear that the referee can recommend our manuscript for publication and appreciate their feedback and suggestions, which we address below.

1. The significantly high $zT \sim 0.7$ in the Ru_2TiSi system compared to the Fe_2VAl (0.3–0.4) full-Heusler system is interesting. How feasible would it be in terms of societal applications, considering the cost and abundance of Ru vs Fe?

Author response: This is, of course, a good point. One strength of the Fe-based Heusler compounds has always been their abundance and cost-effectiveness. Ru is far less abundant than Fe. However, we still do think that our work at least underscores that full-Heusler phases have the potential to achieve higher zT values as well and will hopefully motivate the experimental realization of other semiconducting fH phases. For instance, the Fe-based Heusler compounds Fe_2TiSi , Fe_2TaAl , etc. are predicted to be semiconductors with higher zT than Fe_2VAl yet so far no single-phase synthesis of these phases has been reported. Our study shows that it could be well worthwhile to pursue synthesis efforts of such compounds more wholeheartedly than has been done so far.

2. The authors claim samples $x \leq 0.2$ to be single phase from both XRD and SEM-EDS measurements. There is a minor peak at 40° begins appearing when Hf-content $x > 0.1$, Fig. 1(a). Also, in Figs. 2 (a-c), there are different contrast regions visible. It would be better if authors could provide EDS measurements of these samples at different regions, along with elemental mapping.

Author response: Thank you for pointing this out. We have included the EDX maps and point analysis in response to the 1st reviewer's comment (Fig.R1) and also in the Supplementary Information.

3. Fig. 3(a); Hf-doped ($x \leq 0.2$) samples show profound bipolar behavior compared to the non-doped. Why? Please include explanation in the manuscript.

Author response: By Hf-substitution, we find that the Fermi level shifts slightly closer to the valence band edge. This most likely arises from the fact that the band gap also increases, which requires raising the Fermi level to the valence band top in order to preserve charge neutrality (Hf is isovalent doping). The position of the Fermi level closer to the band edge results in bipolar conduction becoming more visible at lower temperatures since the chemical potential moves quicker toward the conduction band side when E_F is closer to the valence band edge. Bipolar conduction is always observed more strongly in lightly doped narrow-gap semiconductors and diminishes when doping shifts E_F further into the band. Here, while no extrinsic doping occurs, the electronic structure changes (band gap increase) require E_F to move closer to the band edge.

4. Fig. 4(c); the increment in the band gap, E_g is higher an order than that of the reduction in the $E_{VBM} - E_F$ in the Hf-doped samples. However, higher bipolar behaviour can be noticed in the Hf-doped than non-doped. Why? The explanation should be added in the manuscript.

Author response: Please see our comment above. The reason lies in the Fermi level being positioned closer to the band edge and its effect on the temperature dependence of the chemical potential. A larger $E_F - E_{VBM}$ will always result in less pronounced bipolar conduction and is the dominating effect in this case.

5. Did the authors measure Hall carrier concentration and mobility? It would be better to include this data in the manuscript to better explain the transport properties.

Author response: Following the reviewer's suggestion, we measured the Hall effect for these samples, calculated the Hall carrier concentration (n_H), and presented the results in Fig. R4. As expected, n_H remains nearly unchanged up to $x \leq 0.2$, since Hf substitution at the Ti site does not add or remove any electrons (isovalent doping). For multi-phase samples with $x \geq 0.3$, n_H increases due to compositional inhomogeneity and the coexistence of different metallic phases. The corresponding discussion has been incorporated into the main text, and the Hall data are provided in the Supplementary Information.

The Hall mobility (μ_H) shows a behavior in line with transport behavior discussed in the main text. For samples with $x \leq 0.2$, μ_H does not decrease and even gradually increases upon Hf substitution, which we speculate can be attributed to subtle modifications of electron-phonon deformation potentials. Most importantly, this trend further supports the interpretation of orbital-guided electronic-structure engineering in $\text{Ru}_2\text{Ti}_{1-x}\text{Hf}_x\text{Si}$ up to the solubility limit, where Hf substitution does not deteriorate the mobility hole-type charge carriers existing in the Ru-dominated valence band states. Beyond this limit ($x \geq 0.3$), μ_H decreases rapidly, suggesting strong carrier scattering caused by compositional inhomogeneity and the emergence of secondary metallic phases with inherently low mobility.

Figure R4: Hall carrier concentration and Hall mobility of Hf-substituted $\text{Ru}_2\text{Ti}_{1-x}\text{Hf}_x\text{Si}$ ($x = 0.1, 0.2, 0.3$ and 0.4) obtained at 300 K. In the single-phase regime ($x \leq 0.2$), the carrier concentration stays roughly constant, as expected from an isovalent substitution. Surprisingly, the Hall mobility does not decrease and even slightly increases, consistent with the evolution of the weighted mobility.

6. The samples were synthesized at two different laboratories NIMS and TU Wien. It is difficult to understand whether the presented thermoelectric properties belong to the samples synthesized at NIMS or at TU Wien. Also, it would be better if the comparison of the thermoelectric properties of the similar compositions synthesized at the different laboratories will be included in the supporting information, because the followed synthesis route at both the laboratories is different from each other.

Author response: Thank you for pointing out that this was not entirely clear. In fact, all the measured TE properties shown were obtained for the samples synthesized at NIMS. Due to the extremely high melting point of these compounds, rapid cooling during melt synthesis introduces macroscopic cracks in the samples, making them quite brittle. This means that it is difficult to measure as-cast or even annealed samples synthesized through melting. At TUW, we do not have access to a sintering setup. Thus, we only investigated the structural properties of these samples and the TE properties were investigated for the sintered samples at NIMS.

7. Page 2; paragraph, “To investigate the... by XRD (Fig. 2c, d).”; last sentence, “In contrast... by XRD (Fig. 2c, d).” :- please check and correct the figure number.

Author response: Thank you for pointing out the typo. We corrected it to Fig. 2d, e.

References

- Goldstein, J. I.; Newbury, D. E.; Michael, J. R.; Ritchie, N. W.; Scott, J. H. J.; Joy, D. C. *Scanning electron microscopy and X-ray microanalysis*; Springer, 2017.
- Gernert, U.; Berger, D. Depth Resolution and Surface Sensitivity with the Multiple Detection System of a HR-SEM. European Microscopy Congress 2016: Proceedings. 2016; pp 392–393.
- Hinterleitner, B.; Fuchs, P.; Rehak, J.; Garmroudi, F.; Parzer, M.; Waas, M.; Svagera, R.; Steiner, S.; Kishimoto, M.; Moser, R.; others Stoichiometric and off-stoichiometric full Heusler $\text{Fe}_2\text{V}_{1-x}\text{W}_x\text{Al}$ thermoelectric systems. *Physical Review B* **2020**, *102*, 075117.
- Anand, S.; Gurunathan, R.; Soldi, T.; Borgsmiller, L.; Orenstein, R.; Snyder, G. J. Thermoelectric transport of semiconductor full-Heusler VFe_2Al . *Journal of Materials Chemistry C* **2020**, *8*, 10174–10184.
- Hinterleitner, B.; Garmroudi, F.; Reumann, N.; Mori, T.; Bauer, E.; Podloucky, R. The electronic pseudo band gap states and electronic transport of the full-Heusler compound Fe_2VAl . *Journal of Materials Chemistry C* **2021**, *9*, 2073–2085.
- Garmroudi, F.; Parzer, M.; Mori, T.; Bauer, E. Recent progress in Fe- and Ru-based full-Heusler bulk thermoelectrics. *Science and Technology of Advanced Materials* **2025**, 2517537.